# Understanding the Relationship between Depression and Chronic Diseases Such as Diabetes and Hypertension: A Grounded Theory Study

**DOI:** 10.3390/ijerph182212130

**Published:** 2021-11-19

**Authors:** Pablo Alberto Herrera, Solange Campos-Romero, Wilsa Szabo, Pablo Martínez, Viviana Guajardo, Graciela Rojas

**Affiliations:** 1Psychology Department, University of Chile, Santiago 7770204, Chile; pabloherreras@uchile.cl; 2Nursing School, Pontifical Catholic University of Chile, Santiago 7820436, Chile; 3Medicine Faculty, University of Chile, Santiago 8380000, Chile; wilsaszabo@uchile.cl; 4Psychiatry Department, Clinical Hospital, University of Chile, Santiago 8380456, Chile; pablo.alberto.martinez.diaz@usherbrooke.ca (P.M.); viviguajardo@gmail.com (V.G.); graciela.rojas.castillo@gmail.com (G.R.)

**Keywords:** chronic illness, depression, treatment adherence

## Abstract

There is a bi-directional relationship between depression and chronic illnesses such as diabetes and hypertension. This comorbidity is associated with higher mortality risk and diminishes the efficacy of interventions. The specific mechanisms of this mutual influence are still not fully understood, and most intervention protocols address these conditions separately. This study aims to improve our understanding of this relationship. We interviewed 18 patients and 24 health care professionals, focusing on understanding the different ways in which depression and chronic illness could influence each other. Our results show a common cyclical pattern and specific situations where the reported bi-directional relationship does not occur. We discuss the importance of opening a space for the patient’s grief process after the chronic illness diagnosis, managing the demands and stress of the patient’s treatment, and how to adjust the treatment to the different needs and possibilities of each person.

## 1. Introduction

Chronic diseases such as Diabetes and Hypertension (HT) are the leading cause of death and the most frequent focus of care in primary health care centers in Chile and many developed countries [1,2]. They are often presented with comorbidities such as depression [3,4]. This comorbidity is associated with a higher risk of death, a decrease in the efficacy of interventions, and complications in long-term prognosis [2,4,5]. Despite this correlation, the clinical approach to chronic diseases and mental disorders is structured separately in depression, diabetes, and/or cardiovascular health programs. This is inconsistent with evidence showing that depressed patients seeking help at primary health care centers should be treated taking into account multiple comorbidities [2].

A significant limitation is that the specific mechanisms that explain this correlation are still not fully understood [6,7,8]. Researchers suggest that there is high variability among patients regarding the effects of chronic disease on mood and mood on self-care behaviors [6,7]. For example, there is debate about whether psychological interventions to improve depression help in the glycemic control of patients with diabetes, with studies showing positive [8] and negative results [6]. There are even studies that suggest that the management of patients with depression is more effective when they also have diabetes [6]. This problem requires further research, as explaining this correlation (and understanding its exceptions) can be crucial in improving our intervention strategies.

Below, we present different models that explain the relationship between depression and chronic diseases, divided into three large groups.

### 1.1. Mechanisms of Influence: Chronic Diseases → Depression

The first and most important factor that could explain how chronic illness causes or exacerbates depression is the ***burden of suffering from the condition and its impact on people’s quality of life*** [5,6,7,9,10,11]. Being diagnosed as diabetic or chronically ill often has a strong impact on patients beyond the physiological effects of the disease [7]. Most people tend to feel anxiety and depression after the diagnosis because of what it means to them and the uncertainty that looms about their future [9]. Some of the factors that increase this burden are functional deterioration at a physical level and the decrease in physical activity as a result of the deterioration of health [6]; the social isolation associated with the disease [9]; fear of the future and feeling of loss of health due to the diagnosis (9, 10), and guilt for having caused the disease with an unhealthy lifestyle [9].

A factor related to the above is the impact of the chronic condition on the ***sense of self, self-esteem, and locus of control*** of people [6,7,9,10]. The difference between feeling, for example, “a person who has diabetes” vs. “being a diabetic” seems to be key and shows the condition’s relationship with the construction of the subject’s identity. Some authors highlight the influence of the social stigma associated with the chronic condition and how many patients feel a loss or mourning for their previous identity and life, expressing that they feel “trapped in a different life” [9].

The other mechanisms of influence found in the literature point to physiological mechanisms such as the ***effect of blood sugar level on mood*** [6,7,12] and ***structural changes that occur at the brain level as a result of certain chronic conditions*** such as diabetes [13].

### 1.2. Mechanisms of Influence: Depression -→ Chronic Diseases

The main way in which depression accentuates the severity of a chronic condition is by ***hindering people’s self-care behaviors and adherence to medical treatment*** [6,7,8,10,11]. Various explanations are proposed concerning how depression specifically hinders self-care: for example, the pessimism and low self-efficacy of patients with depression leave them with little hope that taking care of themselves and following the treatment will be useful. In addition, the typically lower social support of patients with depression also hinders the development and maintenance of self-care behaviors. Lack of energy, low motivation, and a tendency to use passive coping mechanisms are also important obstacles.

***Depression often causes patients with chronic diseases to do less physical activity in general***, beyond what was stated in the previous paragraph. This is partly due to the fatigue and hesitancy inherent in depression [5] and is also a product of the progressive functional decline experienced by people with long-term depression [10]. These factors would make it more difficult for patients to maintain an active lifestyle and follow treatment, regardless of their motivation or beliefs about its usefulness.

Another related factor is the tendency of people with depression to use ***non-problem-oriented and emotional coping mechanisms, especially emotional eating and binge-eating*** [8,10]. For example, people with depression tend to resolve treatment (and life in general) difficulties in less adaptive ways, making it challenging to maintain self-care behaviors for more extended periods. It has been seen that patients with depression and obesity tend to regulate their negative emotions by eating and often binge-eating [10]. This aggravates chronic diseases such as diabetes and high blood pressure and makes it difficult to maintain the eating habits necessary to compensate for the condition. On the other hand, there is also evidence that this is part of a larger cycle that includes weight gain and loss through multiple diets, which, when they inevitably fail, generate greater damage to the self-esteem and self-efficacy of patients.

***Depression also exacerbates the impact of chronic illness on patients’ quality of life*** [5,8]. Depression makes patients perceive their quality of life as more affected by chronic disease and more pessimistic about the future, which could partly explain the association between depression and patients’ quality of life with chronic diseases. Likewise, depression makes receiving the diagnosis of chronic illness a more devastating and difficult event to cope with.

Other authors suggest that ***depression hinders communication and trust between the patient and the medical team***. Some studies suggest that patients with depression present less satisfaction with the sessions provided by the medical team than do patients with chronic diseases who are not depressed. The health professionals describe the depressed patients as “more difficult” and “less able to cope with the disease” [8]. This means that in certain cases of comorbidity, patient–medical team communication may have negative rather than positive psychological consequences [6].

The results can be understood by considering specific psychological characteristics associated with depression, such as ***the tendency to demand too much of themselves and their high sensitivity to criticism*** [13,14]. Both phenomena have been explained as internalizing negligent or aggressive bonds with past attachment figures [15]. This can lead to (1) expectations of being mistreated or neglected by protective figures, which can make the person with depression engage health professionals with an a priori predisposition of distrust, defensiveness, or confrontation, and interpret “neutral” comments as aggressions, making the bond of trust difficult, and (2) the tendency to neglect themselves [16], which could hinder the self-care behaviors necessary for treatment [17,18].

Finally, some authors suggest that ***antidepressant medications may contribute to the patient’s risk of becoming overweight and developing diabetes***, although studies are lacking to corroborate this association [7,11].

### 1.3. Common Causal Mechanisms

Multiple authors suggest physiological mechanisms. For example, the ***activation and dysregulation of the physiological stress response system*** could influence depression and diabetes, becoming a mediating mechanism for both pathologies [10,11,13]. This could help explain the influence of ***multiple environmental and behavioral factors*** (e.g., poverty and sleep problems) related to this comorbidity [7], as these could generate stress and indirectly contribute to the development of depression and chronic diseases. Other physiological factors that have been associated with both pathologies are the ***alteration of circadian rhythms in both pathologies*** [7,11] and ***aspects of the fetal development environment, which may affect both conditions*** [7]. Other authors have suggested that ***inflammatory mechanisms*** could be a common unspecific etiopathogenic mechanism in depression and some cardiovascular conditions (such as myocardial infarction and its risk factors, including diabetes and hypertension). For example, large amounts of inflammatory cytokines could affect the pancreas and promote the appearance of type 2 diabetes [4].

Finally, some authors propose various ways in which a ***vicious circle could occur between depression and chronic conditions*** (such as hypertension or diabetes) [5,7,8,9]. The literature speaks of two ways in which this circular pattern manifests itself: (a) in depression, healthy behaviors and lifestyle changes required for the treatment of chronic disease are affected, which worsens the condition and quality of life of patients and in turn generates greater depression; (b) Depression generates binge eating and emotional eating as coping mechanisms, which predisposes the patient to obesity, metabolic syndrome, chronic diseases, and difficulties in adherence to diet changes. Likewise, there is frustration with gaining weight and the failure of diets. All of this affects self-esteem and self-efficacy, generating more depression. Regardless of whether this vicious circle occurs, most patients experience both pathologies as related, although for some patients, depression and chronic diseases are independent conditions [9].

Given this context, this study aims to describe and understand the different relationships between depression and chronic diseases and to describe potentially useful therapeutic strategies to intervene in cases with these comorbidities, according to the experience of patients with these comorbidities and health professionals who care for these patients.

## 2. Materials & Methods

### 2.1. Study Design

The study used a qualitative, exploratory, and descriptive design. We incorporated elements of the grounded theory for the relational analysis of the results to generate a substantive theory from the data that expands our understanding of the phenomenon [19,20]. Grounded theory is based on symbolic interactionism, which stresses that the significance of life experiences originates through an interpretation process that is reliant on social interaction [21].

This study is part of a larger project (Fondecyt 1180224) that aims to compare the effectiveness of a collaborative computer-assisted cognitive–behavioral educational and psychological treatment versus enhanced usual care to treat depressed patients with hypertension and/or diabetes in primary care clinics.

### 2.2. Sampling Strategy and Participants

We used the theoretical sampling strategy, collecting new data until meeting the theoretical saturation criterion [22]. Study participants were divided into two groups: (1) people with chronic conditions (hypertension and/or diabetes) and mood problems, and (2) professionals who care for people with these conditions.

(1) Regarding the group of people with chronic conditions and mood problems, the starting criteria were as follows: being over 18 years of age; having received a diagnosis of hypertension and/or diabetes; having a PHQ9 score above nine, or a previous depression diagnosis. People with psychotic symptoms and/or cognitive impairment were excluded. Most participants were currently on medication for depression, but not all were currently on a psychotherapy treatment.

We conducted 18 interviews with people between 18 and 78 years of age (M = 60 years), 14 women and five men. All attended Public Health Centers (where approximately 80% of the Chilean population attend, which corresponds to the lower salaries) and lived in the Chilean capital. Seven of them were housewives, two were retired, two were professionals, one studied at the university, and the remaining six had low-qualification jobs. All had been diagnosed with depression, but three were already discharged with no current symptoms. Their average PHQ9 score was 13 points (moderate symptoms). Fourteen had hypertension, and 11 had diabetes (seven of the 18 had both conditions).

Subsequently, to clarify certain hypotheses arising from the analysis, we analyzed 22 additional interviews (bringing the total to 40 analyzed interviews) that met the same inclusion criteria and had been part of a previous study [23]. Of these 22 additional interviews, 4 were male and 18 were female. The age range was from 33 to 71 years (average 54). Twelve were treated in the public health system, and 10 in the private system.

(2) Regarding the group of professionals, we carried out five focus groups in which a total of 27 people participated (20 women and seven men). Of these, 10 were doctors (eight of them doing their psychiatry specialization), seven were nurses, eight were psychologists, one was a nutritionist, and one was a social worker. Their years of experience with chronic patients ranged from five months to 29 years (M = 7.9 years). Fifteen of them had training in depression and/or chronic diseases. All attended in institutions of the Chilean public health system: 24 in primary care clinics and three in hospitals. In addition to these focus groups, we individually interviewed one psychologist and one nurse, both experts in caring for patients with chronic conditions. For these interviews, we used the same thematic script of the focus groups. We interviewed them individually for scheduling reasons.

The inclusion criteria were as follows: regular care of patients with depression + hypertension and/or diabetes comorbidity; at least one year of clinical experience. Regular care was defined as working in the cardiovascular and mood programs, using the Chilean health protocols for those conditions.

### 2.3. Procedure

To recruit participants, we contacted professionals who worked in public health institutions in Santiago de Chile (one hospital and 4 primary care centers). Within the framework of the Fondecyt 1180224 project, we requested authorization from the directors of the respective services to invite patients and health professionals to participate in this research. Most patients were invited to participate in the waiting rooms of the primary care centers. Complementarily, we contacted patients of the main researcher of the Fondecyt project (private sector) and invited them to participate. The focus groups were carried out in the Clinical Hospital of the University of Chile and in the primary care centers where the participants worked, in a room set up for a small group of people. No participant received financial compensation for their participation in the study. According to their preference, the patient interviews were carried out in a private psychiatric consultation in the office where each patient was treated or at the patient’s home.

The research team that conducted the interviews comprised the first three authors of this study: two psychologists and a nurse. At least two facilitators were present in each focus group. An interviewer conducted the interviews (except for one session in which two interviewers were present). The focus groups lasted between 60 and 80 min. The interviews lasted approximately 45–60 min, were transcribed verbatim, and participants were identified with a pseudonym to protect their confidentiality. The rationale for using individual interviews for patients and mostly focus groups for health professionals was that the patient interviews included exploration of their emotions and experiences with difficult topics, which required a private, secure space. Conversely, health professionals were asked about their professional experiences without inquiring about personal or sensitive topics.

### 2.4. Data Collection Instruments

The present study used two types of instruments: a screening instrument and data collection instruments. They are described below.

#### 2.4.1. Diagnostic Instrument for Patients

Patient Health Questionnaire-9 item (PHQ-9): a self-administered nine-item instrument designed to assess the severity of depressive symptoms. This instrument was validated for the Chilean primary health care contexts [24].

#### 2.4.2. Data Collection Instruments for Patients and Health Professionals

A patient questionnaire with sociodemographic and health information: used to collect information on sociodemographic variables, health status, medications, etc.

Semistructured interview for patients. We used the following thematic script: reaction to the diagnosis (e.g., “how was it for you when you got the diagnosis for your diabetes?”); experience with indicated medical treatments (e.g., “could you tell me a little bit about how it has been for you having to take the medication and trying to adhere to the diet and exercise indications?”); easy and difficult situations in the management of the chronic condition and mood problems (e.g., “what was easiest and most difficult for you regarding adhering to the treatment for your hypertension?”); experience with comorbidity (e.g., “what happens with your hypertension treatment when you have been depressed?”, “do you see any relationship between your diabetes and depression?”); positive and difficult situations with treating professionals (e.g., “how has been your experience with the health professionals?”); motivations for and against following the treatment (e.g., “could you tell me more about the situations when it’s most difficult for you to adhere with the treatment? What do you think makes it difficult, in that moment?”).

Focus groups and interviews with professionals. In both cases, we used the following thematic script: effective and ineffective experiences and learning from clinical practice with patients with depression/chronic condition comorbidity; situations that made it difficult for patients to adhere to treatment; characteristics of patients that are more “easy” and “difficult” to work with, as well as those who present “good” and “bad” therapeutic results; evaluation of current treatment protocols; ideas and lessons learned about possible improvements to current protocols.

### 2.5. Data Analysis

The data analysis was carried out jointly by the first two authors of this article (a nurse with decades of experience designing and applying interventions to help people with chronic diseases; a psychologist with more than a decade of experience researching the difficulties and ambivalence that patients face when adhering to treatments). At the beginning, the analysis included an explicit positioning or “bracketing”. The constant triangulation of the analysis (intersubjective validation) made it possible to ensure the quality and rigor of the results [25].

In the case of the professionals, we began by carrying out the open coding of the interviews. We then adjusted the details of the thematic script for the focus groups and continued the recursive process of analysis and collection of information, typical of grounded theory, until saturating the categories of analysis.

Subsequently, we began collecting data from the group of patients. We jointly analyzed the first interviews. Based on these preliminary results, we continued collecting information, looking for new interviews that would allow us to explore the hypotheses that emerged from the initial data.

To organize the results, we decided to present the analyses from a relational, not merely descriptive, perspective: we showed all our results regarding the possible relationships between chronic conditions and the patients’ mood (we chose to speak of “mood” because “depression” was too restrictive, especially for analyzing the experiences of the patients). For this reason, we integrated the results of the group of professionals and patients.

Finally, we carried out a selective analysis in which we sought to propose a model that integrates the central points of the results.

Regarding methodological rigor, we followed Thomas and Magilvy’s recommendations [26]. For the credibility criteria, we used the member check procedure with two professionals and one patient. The PhDs and research experience on chronic illness of the main authors also contributed to the credibility of the results. For the transferability criteria, we provide a detailed description of the participants and their context. We also included health professionals with experience working with patients in large cities and rural centers. For the dependability criteria, we triangulated the analysis between the two main authors and presented a detailed description of our procedure. Additionally, the authors can be contacted to ask for detailed interview guides and orientation on replicating the study. For the confirmability criteria, we took detailed notes after every interview and regularly discussed our assumptions and biases. The different professions of the two main authors helped us maintain a constant self-critical attitude.

## 3. Results

Below, we present the main relational results of the study: (1) the common vicious circle that occurs between both conditions and the contextual factors that influence it; (2) the exceptions in which this relationship does not appear; (3) different suggested and experienced therapeutic strategies.

### 3.1. Vicious Circle: Bidirectional Relationship between Mood Problems and Difficulties with the Management of Chronic Disease

In most cases, we identified a bidirectional relationship between both types of pathology, in which chronic disease negatively influences mood, and depression negatively influences the chronic disease (Figure 1). This easily becomes a vicious circle in which the patients, by not taking care of themselves, worsen their chronic condition, which generates more hopelessness about the future and causes the treatment to be neglected further. In these cases, patients are not able to maintain their adherence to treatment, which in turn makes them feel very guilty and self-critical, deepening their depression. Now we will describe this in more detail.

#### 3.1.1. The Ways Chronic Disease Affects Mood

##### Chronic Illness Produces Grief and Hopelessness about the Future

In some cases, chronic disease was experienced as a significant loss for the person (either because of what the condition means in itself or because of the indicated treatment). Faced with this loss, the person imagines a hopeless future. This can manifest as sadness, an angry refusal to follow treatment, or a state of shock in which the person is paralyzed without knowing what to do. These different reactions can be understood as different stages or responses to grief. Losses can be diverse: sometimes related to the person’s self-image (feeling old, sick, different); other times as fear of the terrible and inevitable consequences that the person imagines; other times as anxiety about treatment, which may involve changes in habits that affect important aspects of life and limit well-being.


*“I have a vision problem. Suddenly, I say that it’s best if I’m no longer alive, because I’m going to be a burden. If I go blind, I’m going to be a burden and I don’t want to be.”*
(*E4)*


*[On how he reacted when his diabetes was confirmed] “... I didn’t want it. I didn’t even want to see a doctor. I had no idea that I had to be with a nutritionist or anything. So, it bothered me, and I was scared about what to do. So that’s how I spent many years. (...) Suddenly, I was taking [the medications] and other times I was not taking them. I have to be very honest about that, because I was angry, upset... Why did I get sick like that?”*

*(E9)*


##### Chronic Illness Causes Discomfort and Stress

Some people found their mood affected by the discomfort of their chronic disease or the inconvenience of the following treatment. This reflects situations in which the complications of the disease or the costs of including its treatments in daily routines generate constant stress and irritation in the person’s life, affecting their mood. This category differs from the previous one because not every reaction of irritation and stress from treatment and chronic illness is explained as unresolved grief.


*“... But one doesn’t have time. Vegetables—do not eat this vegetable, do not eat the other vegetable. In the end, I don’t eat vegetables, I just like lettuce... tomatoes, generally, tomatoes. But I can’t eat tomatoes either because tomatoes are too sweet. Corn is also sweet. So, in the end, what does one eat?”*

*(E6)*


#### 3.1.2. The Ways Mood Problems Affect the Evolution of Chronic Diseases and/or Adherence to Their Treatment

##### Depression Makes You Feel It Is Pointless to Take Care of Yourself

One of the characteristics of depression is the feeling of hopelessness for the future. In these cases, when the person sees an inescapably negative future, the purpose of making an effort to change health and self-care habits is lost, making it challenging to continue the treatment of chronic disease. In these cases, the cognitive biases typical of depression [27] lead the chronic condition to be perceived and experienced as even more catastrophic, rendering the person unable to cope with the disease.


*“If one day you see everything black and you see that it will not change at all, that your life has no meaning. It does not matter to you if you dialyze yourself or not, or if you follow the instructions or not, if in the end what you want is to stop living”.*

*(EntProf2)*


##### Maintaining a Healthy Lifestyle Requires Too Much Effort for the Depressed Person

Another characteristic of many depressions is the lack of vital energy. In these cases, even if the person does not feel completely hopeless regarding their future, they lack sufficient energy to mobilize and maintain their self-care habits. This difficulty is often underestimated or not perceived by health professionals and people close to the patient. They expect patients to make an effort they are not able to make.


*“You still feel lonely because nobody understands you in the end. Because they are judging you because you are fat. For example, my mom didn’t believe me when I told her that I suffered from anxiety. And she told me: ‘you must stop eating so much’. And I tell her: ‘ah, but it’s not that easy’ (...) Besides, the endocrinologist says things a bit harsher. They say: ‘now, take this…do it’”.*

*(Ent5)*


##### Patients with Depression Are Distrustful of Authority Figures, Making It Difficult to Collaborate with Health Professionals

In several cases, patients with depression reported feeling fear, distrust, or even anger towards the professionals who cared for them. This was attributed to aggressive treatment, perceiving medical indications as challenges, and criticisms that could even be malicious. All the above generated a rupture in the collaborative bond. This can sometimes be due to the excessively directive or aggressive style of some professionals. However, it is also due to bad experiences of patients with other professionals or attachment figures from their history. This can lead the person with depression to meet the treating professional with an a priori predisposition of distrust, defensiveness, or confrontation. Patient mistrust of health care providers and the health care system has been reported not only in depressed patients [28], but our results suggest that depressed individuals could show traits that make them more mistrustful than other kinds of patients.


*“Patients who do not adhere to treatment or sessions or whatever, it is because they have had bad experiences, either with doctors, with the health center, the hospital or whoever. Then with those bad experiences like, I don’t know … As they already begin to reject the consultation a little.”*

*(GF4)*



*[Why did you make the decision to stop seeing your doctor?] “I didn’t want to live my whole life taking pills. I don’t want to live my whole life thinking ‘NO! YOU CAN’T EAT THIS, LEAVE IT THERE!’ Because many people are like that. Even my mom: “you can’t eat that.” It makes me sick (nervous laugh), so I didn’t want to live like this, that is, I preferred to die but not live like this. Because it’s not easy”.*

*(Ent15)*


##### Self-Neglect Patterns Hinder Self-Care Behaviors

In several of our interviewees, we observed a tendency to demand too much of themselves and live focused on others’ expectations and needs. This made it difficult for them to dedicate time to their self-care and maintenance of healthy habits.


*(...) “I spent many years worrying more than one hundred percent about him and not caring about myself. (…) I worried about him taking the exams, about his meals, about everything. And I... I forgot the date of... to go to the nurse, to go to the doctor. Then I realized… I failed to go to the doctor’s appointment.”*

*(Ent9)*


##### The Body Manifests the Consequences of Emotional Distress

Several patients explained their physical problems as an effect of emotional distress (stress, depression, guilt, etc.). These mood problems could have triggered the onset of chronic disease or the exacerbation of symptoms.


*“I told him I have triglycerides sky high and diabetes sky high, and it is because I was not calm. (...) I had no time for anything. My day was nothing. I lacked hours a day to continue doing things. (...) So, I think that neuroses are involved a lot. Yes, because I get nervous. And everything raises your sugar”.*

*(Ent7)*


#### 3.1.3. Contextual Factors

The vicious circle we have just described does not occur in a vacuum. It can be facilitated or hindered by environmental conditions that can differ in otherwise similar patients who present these health conditions.

We observed three characteristics of the patients that are relevant for the development of this dynamic: (1) the degree of self-demand and self-criticism versus self-compassion that they present, (2) their basal mood tone (e.g., if they are used to living with dysthymia they can continue their treatment despite their mood problems), (3) their history of relationships with attachment and authority figures, and (4) their previous healthy or unhealthy habits.

Regarding the characteristics of the context, we observed four main elements: (1) the therapeutic bond with the health professionals (collaborative vs. unproductive), (2) the level of family support (facilitating and supportive vs. critical and demanding), (3) the family history and beliefs regarding chronic diseases (e.g., if the chronic condition is seen as “normal”, it may be easier to assume), and (4) the availability of community networks.

These contextual factors can be better understood by analyzing the exceptions to the vicious circle.

### 3.2. Exceptions

This vicious circle did not occur in all cases, as there were several ***exceptions*** in which the chronic condition did not worsen the patient’s mood, or the depression did not worsen the chronic condition.

#### 3.2.1. Cases in Which the Chronic Disease Does Not Adversely Affect Mood

##### Chronic Disease Is Normalized or Does Not Imply a Big Change in Daily Habits

In some cases, awareness of chronic disease did not cause shock or mourning for the person since it was expected or integrated into their identity and daily life (either due to family history or because it is more expected with advanced age, etc.). In other cases, the disease did not imply a great change in health habits (for example, there were already other people in the family with a chronic condition who eat healthily). In all these cases, the chronic disease did not have a dramatic significance or create a major change in self-image or lifestyle. Therefore, the disease did not seem to affect the person’s mood.


*“But it happened to me after my 60th birthday. That is why I say that it has not been an issue for me.”*

*(E4)*


##### Treatment as an Opportunity to Care for and Love Myself

Some people signified their diagnosis and treatment experience as a self-care experience, creating a positive feeling (spontaneously or thanks to therapeutic support). The chronic illness appeared as an alarm signal for the person, a sign that they must take care of themselves, love themselves, and prioritize themselves instead of spending all their energy caring for others or demanding too much of themselves.


*“My life was always about home, son, work, son, work, home, work... so it was very busy, and I never worried about myself (...) I started there and finally said ‘no’, now I’m going to start caring about me. They are all doing great, I have no greater responsibility, so I’m going to start with my own affairs. And that’s when I started to look out for myself.”*

*(E2)*


##### Self-Regulation Ability Protects Mood

In these cases, the chronic disease did lead to relevant changes in the person’s lifestyle and habits. However, these changes failed to negatively affect the mood because the person managed to integrate them into their life context, adapting the treatment to the possibilities and conditions of their life without over-demanding or blaming themself for the things they cannot accomplish. The person used their self-management and self-regulation abilities, helping to ensure that the treatment of the chronic disease is not such a burden.


*“She [her daughter] arrives with delicious things, …a lemon pie, a kuchen, … a milk cake, prepared by her. And she says to me: ‘look mom, see what I brought’. But if you are going to give me something, I say, me, give me only a bit, the size of a box of matches. And yes, I accept a bit of cake, so I don’t hurt her feelings. I say thank you, because it doesn’t complicate life for me, because it’s not so much and it’s not every day. What else can I do?”*

*(E8)*


#### 3.2.2. Cases in Which Mood Problems Do Not Affect Chronic Diseases

##### A Self-Disciplined Person with a Long-Standing “Normalized” Depressive State Continues Her Treatment despite Feeling Bad

Some people reported that their depression is “chronic” or long-standing, so it is not an acute depressive episode but almost a stable part of their life (such as recurring depression or dysthymia). In this situation, depression does not mean a major change in their life. In general, these people have a very strong self-demanding personality trait. In these cases, even if they feel bad or do not feel like doing anything, it is unthinkable for them to stop fulfilling their obligations and tasks (and therefore follow treatment, which they see as a duty).


*“I always take the medicine. Even if I have all the problems I have, but I always do the same... I have never stopped taking the medicines. (...) Calm or desperate, I don’t know. But I always take the medicine. I am not irresponsible in taking the pills.”*

*(Ent10)*


### 3.3. Suggested and Experienced Therapeutic Strategies

#### 3.3.1. Acknowledge the Loss and Grief Process

This refers to the patients’ need to embrace the feelings of sadness and loss that are often generated after diagnosing a chronic disease. Space needs to be given to these emotions to allow their elaboration and integration into the person’s self-image and vital narrative. The first step for health professionals is to recognize that there is a grieving process, and not focus the entire intervention on giving suggestions and directions.


*“... the idea of helping him so that that person can acknowledge that ‘loss’, regarding his image of himself, his lifestyle and everything in a way that he can continue living with it. Give the person time to experience the grief and accompany him in that and not expect the person to accept it immediately, knowing that grief also has its processes.”*

*(EntProf2)*


#### 3.3.2. Promote Empowerment and Hope

This refers to the ability of professionals to convey hope to patients, especially when they see a hopeless future due to their chronic disease. One of the ways to achieve this is by giving specific information related to patients’ fears, showing concrete actions to prevent these feared outcomes. In addition, professionals can instill hope by prescribing an achievable treatment plan according to the reality and context of the person.


*“She became depressed thinking that complications could lead to amputations, loss of sight (...). She had to first learn to live with her disease and to be able to cope with all the changes associated with it, such as eating, mealtimes, check-ups, and then she realized that if she kept her condition stable, she would be fine, she wouldn’t have complications.”*

*(GF1)*


#### 3.3.3. Support the Patient’s Context (Family, Community) so That the Person Requires Less Effort to Follow Their Treatment

This refers to the idea that interventions should integrate the meaningful contexts of the patients (family, work, community) and not solely focus on the “chronic patient”. This is especially important because changing lifestyle habits requires energy and perseverance, and depression makes any effort more difficult for patients. In some cases, family members try to support people with chronic disease by putting pressure on them, challenging them, and insisting that they must adhere to treatment. This can be counterproductive, generating a reaction contrary to what was expected.


*“My mother says that it is a disease for people who are old, not for the young. And besides, I have a lot of family who are diabetic, a lot. So, my mom always tells me: look, you want to end up like them? [family members with diabetes]. (…) since I don’t take care of myself and all those things. And my mom is nagging. (...) So when my mom says those things to me, she hurts me a lot. Like: do you want to look like your aunt? (…) I start to cry.”*

*(Ent5)*



*“So, my husband is involved in my [treatment] plan. He also helps me jog. He puts the TV on for me, he puts on music so that I don’t get bored. (...) Since there are not many of us at home, there is awareness on the part of my husband, which is the main thing. Sometimes, when my son wants to eat food that’s bad for me, he says ‘mommy, don’t wait for me with food, because I’ve already had lunch, or I’ve already eaten’.”*

*(Ent8)*


## 4. Conclusions

Our results showed a common vicious circle between chronic conditions and depression. However, there were exceptions in which the chronic condition did not negatively affect the patient’s mood and in which mood problems did not worsen the chronic condition. We also presented the main therapeutic strategies suggested by the patients and health professionals we interviewed. In this final section, we highlight three main conclusions, emphasizing their possible clinical utility. Each of them is accompanied by a discussion of the relevant literature.

### 4.1. Acknowledge and Support the Patient’s Grieving Process

There was consensus between patients and health professionals that chronic conditions often produce grief and hopelessness. Persons diagnosed with a chronic condition face multiple losses: they must stop certain behaviors that can bring them pleasure or relief, they must dedicate extra effort to things that used to be easy, and they may face losses in their physical capacities with important aspects of their future suddenly threatened. This means a grief process, a concept that does not usually stand out in the literature (or in our health professionals’ training, which focuses on acute and “reparable” diseases). Concepts such as “hopelessness” and “change at the identity level” appear in the literature e.g., [6,10], acknowledging that the burden of the condition and its impact on the sense of self could lead to depression [6,9]. However, hopelessness is sometimes understood as an irrational or pathogenic cognition in the person, motivating the professional to change those beliefs or focus on the “healthier” ideas and attitudes of the patient. Doing this with a grieving patient is anti-therapeutic since the patient needs a space to express and process their difficult feelings [29,30]. Furthermore, focusing on self-care habits without acknowledging and supporting the grief process can worsen the problem: weakening the patient–professional bond, making the patient feel even more guilty [9,31], and generating additional stress with demands that the weakened person is not able to meet [18].

What would a healthy grieving process look like in patients diagnosed with chronic diseases? This is also not typically highlighted in the literature on the stages and management of grief e.g., [30], and our results can only suggest certain ideas that require further study. With these limitations, we suggest keeping three principles in mind: (1) welcome and make visible the possible feelings of loss (grief, anger, hopelessness, fear, etc.) triggered by the diagnosis of chronic disease; (2) facilitate and enhance social support (empathetic rather than critical or demanding) for the grieving patient; (3) after the above, enhance self-efficacy and hope, explicitly associating the patient’s self-care behaviors with the future consequences of the disease. All of this could help patients apply healthy coping strategies, as observed in a pioneering study by Grulke et al. [32] in patients with leukemia.

### 4.2. Support the Patient’s Management of Treatment-Related Stress and Demands 

Even if a grieving process is not generated, almost all our participants acknowledged the great burden and added stress that chronic patients face because of the treatment and consequences of the condition. This burden appears in the literature e.g., [7] and is heavier for a person suffering from depression and its concomitant energy depletion. When patients perceive that the demands of treatment and the demands of the environment are greater than their coping capacities, they feel stress [33,34]. If this occurs, there is a clear risk of worsening the patient’s chronic illness and depression [7,10,11,13].

The susceptibility of patients with chronic illnesses to stress is accentuated by the self-criticism observed in many patients with depression [13]. This trend is also associated with hypersensitivity to the criticism or demands of others [14], as the recommendations of the health professionals can be perceived as scolding or punishment by the patients. This process can facilitate a vicious circle in which the patient feels bad about herself and is criticized by the health professional, weakening the patient–caregiver bond and worsening adherence. This generates, in turn, a reaction of frustration and burnout on the part of an increasingly challenged health professional, which worsens the situation [17,18].

However, our results and some recent studies suggest a way to break out of this vicious circle and help patients improve their self-care behaviors in an empathic and sustainable way over time. For instance, it has been suggested that caregivers accept and explore patients’ difficulties with treatments and self-care tasks instead of “pushing” adherence [17,23]. Complementarily, the “Minimally Disruptive Medicine” [35] recognizes the burden of following the treatments, changing patients’ lifestyle, and the capacities of each patient to take care of themselves (while continuing to work, take care of their family, among other roles). It proposes that attention focused on the individual patient be designed so that it exerts the least possible burden on their life.

### 4.3. The Importance of “Exceptions” and Tailored Treatment

Our study started from the premise that there is a relationship “greater than chance” between chronic illness and depression. Our results are mainly focused on better understanding why and how this relationship occurs. However, there are cases within our sample in which this relationship does not exist (see Section 3.2: “exceptions”). These cases are not highlighted in correlational studies or intervention policies that aim to apply a single treatment that helps the largest number of patients. As we have seen, the medical treatment of chronic disease for some patients does not imply a great change and does not generate any grief. In other cases, the patients manage to self-regulate and manage the “burden” of the disease well. There are even special cases where the chronic disease diagnosis does not generate depression, serving instead as a trigger to raise awareness that it is time to take care of themselves and stop caring only for others, generating positive effects on their mood.

All these exceptional cases highlight the importance of a “tailored treatment” that recognizes each patient’s characteristics, context, and values [36,37,38,39]. At the international and Chilean level, multiple interventions have been developed in recent years that follow these principles and take advantage of the opportunities offered by telemedicine for a more individualized monitoring of each patient [40,41,42]. Studying these “exceptions” can also serve to illuminate future research that helps us continue to learn from our patients and their successful coping strategies.

### 4.4. Final Words

The limitations of the study must be considered when interpreting the results. First, we collected the experiences of a relatively small and culturally homogeneous group of people. Likewise, our investigation of the emotions and implicit meanings of the patients was limited by their capacity for self-report and by the possible difficulties of expressing themselves before an unknown interviewer in a research context. We believe that future studies can explore the situation of patients and professionals in other cultural contexts, expanding the sample and considering the exceptional situations that we observed but that have not been previously highlighted in the literature. In addition, subsequent studies may continue to explore the resources and successful strategies that professionals and patients report. Finally, our results do not account for the differences in perspectives between patients and health professionals. We made this decision because our focus was on the different potential explanations for the correlation between depression and chronic conditions, irrespective of the data source. However, future studies could differentiate between the experiences of different subgroups, for example, comparing patients with different personality traits or types of mood disorders.

At the clinical level, we believe that it is worth reinforcing the therapeutic strategies that emerged from our study: the empathic and supportive approach, centered on the person, is more valuable than assuming a position of critical authority, detached from the contexts and concrete possibilities of our patients and users.

## Figures and Tables

**Figure 1 ijerph-18-12130-f001:**
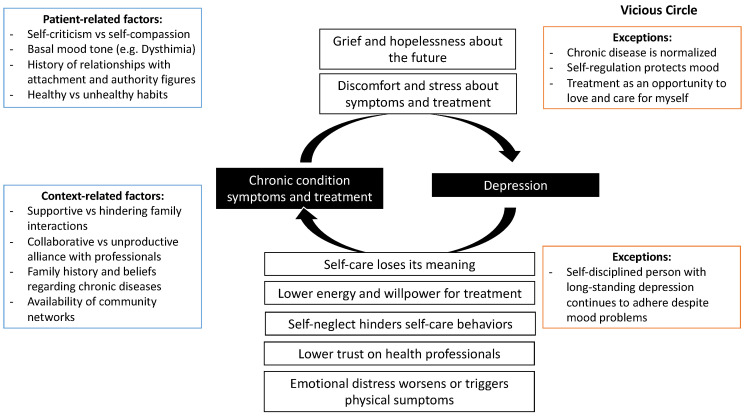
Common vicious circle between chronic conditions and depression.

## Data Availability

The data presented in this study are available on request from the corresponding author. The data are not publicly available because the whole interview transcriptions contain information that could identify the participants.

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
