# Peer review of "Understanding the Relationship between Depression and Chronic Diseases Such as Diabetes and Hypertension: A Grounded Theory Study"

_ijerph, 2021, doi:10.3390/ijerph182212130_

Round 1

Reviewer 1 Report

  • Lines 46-48 repeat one another. Line 48 can be removed.
  • The aims seem to come prematurely in the introduction. It would read better to build up to the rationale and report the aims at the end of the introduction. This would then flow nicely into the methods section.
  • The authors don’t give a rationale for the use of a grounded theory methodology, and this should be added within the paper.
  • It’s not mentioned where participants were recruited from (either patients nor professionals).
  • The inclusion criteria for professionals is listed to include ‘regular care of patients with depression + hypertention and/or diabetes’ but it’s not stated how ‘regular care’ was defined.
  • What was the rationale for using individual interviews for the patient group, but focus groups for the professional group?
  • It would benefit to utilize subheadings within the ‘data collection instruments’ section.
  • It would aid to include some examples of questions used in the interviews, or an excerpt of the interview schedule.
  • In the data analysis section, whilst the authors mention that analysis included explicit positioning/bracketing, it would add to mention the professional background of the researchers that completed the analysis.
  • It would strengthen the paper to include a section on rigor which details the various methods and strategies used to enhance rigor. There are many mentioned in the paper, and this is certainly a strength of the research. However, these aren’t discussed in much detail although are critical to the quality of the research.
  • The figure included in the results section needs a caption (i.e. Figure 1. A model of… ). ‘Figure 1’ can then be used throughout the results to refer the reader back to the model when useful to do so.
  • The results appear quite cluttered. Formatting could be improved by making the headings clearer, particularly the 4th level headings. This would help with readability and navigating through the results section.
  • Within the results, participant quotes are just placed at the end of sections following commentary. Whilst not essential to understanding the paper, it would improve the quality to intergrate these within the commentary.
  • Another limitation not currently considered or mentioned in the paper is the failure to account for differences in perspectives between patients and professionals, and also between interprofessional groups. This is a common flaw of many qualitative approaches, and is not unique to this study. However it is important to acknowledge that the results reflect a collated consensus across patients and various professional groups.
  • Whilst the discussions within the conclusion do draw upon the results in places, a lot of the conclusions drawn are linked back to the wider literature field, but less so the research findings. As such, the discussion section does read as more of a commentary on the literature base, with some subsequent reflection on the results here and there, rather than a primary comprehensive exploration of the results which is then, secondarily, linked back to the wider literature. In other words, it reads as though the primary focus is on the literature, with reference to the results section to support/contrast this, whereas the primary focus should be on the results of the study, with reference to the literature to support/contrast and explain these. More explicit reference to the results of the current study is needed. 

Author Response

  • Lines 46-48 repeat one another. Line 48 can be removed.

We removed the repeated line.

  • The aims seem to come prematurely in the introduction. It would read better to build up to the rationale and report the aims at the end of the introduction. This would then flow nicely into the methods section.

We changed the location of that paragraph. Now it's at the end of the introduction.

  • The authors don’t give a rationale for the use of a grounded theory methodology, and this should be added within the paper.

We added a rationale.

  • It’s not mentioned where participants were recruited from (either patients nor professionals).

There was some information already in the paper and we added more information in the 2.2 section.

  • The inclusion criteria for professionals is listed to include ‘regular care of patients with depression + hypertention and/or diabetes’ but it’s not stated how ‘regular care’ was defined.

We added a definition in section 2.1.

  • What was the rationale for using individual interviews for the patient group, but focus groups for the professional group?

We added an explanation in section 2.2.

  • It would benefit to utilize subheadings within the ‘data collection instruments’ section.

There was a problem with the formatting in the change between our original version and the version that was sent to reviewers. We made many changes to correct this.

  • It would aid to include some examples of questions used in the interviews, or an excerpt of the interview schedule.

We added at least one example question to each of the topics.

  • In the data analysis section, whilst the authors mention that analysis included explicit positioning/bracketing, it would add to mention the professional background of the researchers that completed the analysis.

We added this

  • It would strengthen the paper to include a section on rigor which details the various methods and strategies used to enhance rigor. There are many mentioned in the paper, and this is certainly a strength of the research. However, these aren’t discussed in much detail although are critical to the quality of the research.

We added a section detailing this.

  • The figure included in the results section needs a caption (i.e. Figure 1. A model of… ). ‘Figure 1’ can then be used throughout the results to refer the reader back to the model when useful to do so.

We added this. 

  • The results appear quite cluttered. Formatting could be improved by making the headings clearer, particularly the 4thlevel headings. This would help with readability and navigating through the results section.

We corrected this problem that arose in the process of changing the format after we sent our document.

  • Within the results, participant quotes are just placed at the end of sections following commentary. Whilst not essential to understanding the paper, it would improve the quality to intergrate these within the commentary.

We decided not to integrate the quotes with the commentary in each of the sections of the results. If it's necessary we will integrate them, but for us it's easier and more familiar to present them like they are now. Perhaps it's a question of styles.

  • Another limitation not currently considered or mentioned in the paper is the failure to account for differences in perspectives between patients and professionals, and also between interprofessional groups. This is a common flaw of many qualitative approaches, and is not unique to this study. However it is important to acknowledge that the results reflect a collated consensus across patients and various professional groups.

We added this as a limitation

  • Whilst the discussions within the conclusion do draw upon the results in places, a lot of the conclusions drawn are linked back to the wider literature field, but less so the research findings. As such, the discussion section does read as more of a commentary on the literature base, with some subsequent reflection on the results here and there, rather than a primary comprehensive exploration of the results which is then, secondarily, linked back to the wider literature. In other words, it reads as though the primary focus is on the literature, with reference to the results section to support/contrast this, whereas the primary focus should be on the results of the study, with reference to the literature to support/contrast and explain these. More explicit reference to the results of the current study is needed. 

We made explicit how the conclusions arise from the results. We hope it's clearer now.

Reviewer 2 Report

The study by Herrera et al takes a different approach to understanding the interaction between depression and chronic diseases. This is a relevant topic given the growing prevalence worldwide of type 2 diabetes and cardiopulmonary diseases, both stemming from long-term poor diet. The authors make several recommendations to clinicians and other caregivers who may need to navigate sensitive topics in order to improve patient compliance with lifestyle changes and medications, while also providing perspective into why some patients avoid treatment and help for life-threatening ailments. Overall, the paper is informative, but will benefit from some edits and expansion of interpreting their results. Additionally, the idea of a circular relationship between depression and diabetes is not new, but the paper could benefit from a stronger recognition of the existing literature, which could help highlight why their approach is filling a gap.

Introduction 

Line 39-clarify what you mean in that final sentence, it's very confusing.

Line 46-these final two sentences say basically the same thing, please consolidate into a single point.

Check the formating of your headings throughout this section, they are inconsistent. And you don't label a third point, despite indicating that there were 3 points to this section.

Line 130-under the "common causal mechanisms" section, this area would benefit from including information on how heavily different inflammatory mechanisms play into both depression and chronic conditions.

The end of the introduction is missing a summary statement/hypothesis that leads into the purpose of the study and briefly how you addressed this question.

Methods

Line 168-please include information in this section on whether any participants in the study were currently on medication for depression or seeking counseling.

For general formatting throughout the paper, any number under 10 and at the beginning of a sentence should be spelled out, such as in lines 179, 189, 191, and many others.

Line 191- It is unclear why the authors included individual interviews with the psychologist and nurse. Please provide clarity on this.

Results

Consider revising your figure to include the "exceptions to your theory.

Line 280-HT was never defined.

In general, it feels that the authors endorse that chronic illness is leading to depression, can this please be elaborated in and provide more examples from the literature that support this.

Section 3.1.2.3-Do patients with chronic health problems, but no depression, also have mistrust of Healthcare providers? Please include some acknowledgement of this as either providing data or as a limitation of the study.

Author Response

The study by Herrera et al takes a different approach to understanding the interaction between depression and chronic diseases. This is a relevant topic given the growing prevalence worldwide of type 2 diabetes and cardiopulmonary diseases, both stemming from long-term poor diet. The authors make several recommendations to clinicians and other caregivers who may need to navigate sensitive topics in order to improve patient compliance with lifestyle changes and medications, while also providing perspective into why some patients avoid treatment and help for life-threatening ailments. Overall, the paper is informative, but will benefit from some edits and expansion of interpreting their results. Additionally, the idea of a circular relationship between depression and diabetes is not new, but the paper could benefit from a stronger recognition of the existing literature, which could help highlight why their approach is filling a gap.

We added more information about the gap or research problem that justifies our study

Introduction 

Line 39-clarify what you mean in that final sentence, it's very confusing.

We changed that line

Line 46-these final two sentences say basically the same thing, please consolidate into a single point.

We did that. Thanks for pointing that mistake

Check the formating of your headings throughout this section, they are inconsistent. And you don't label a third point, despite indicating that there were 3 points to this section.

There were many formatting problems in the transformation of our original document to the version sent to reviewers. We corrected them

Line 130-under the "common causal mechanisms" section, this area would benefit from including information on how heavily different inflammatory mechanisms play into both depression and chronic conditions.

We added information about that.

The end of the introduction is missing a summary statement/hypothesis that leads into the purpose of the study and briefly how you addressed this question.

We placed a paragraph there with that purpose

Methods

Line 168-please include information in this section on whether any participants in the study were currently on medication for depression or seeking counseling.

We added this information, although we didn't collect that info in a formal way (only as part of the interviews)

For general formatting throughout the paper, any number under 10 and at the beginning of a sentence should be spelled out, such as in lines 179, 189, 191, and many others.

Corrected

Line 191- It is unclear why the authors included individual interviews with the psychologist and nurse. Please provide clarity on this.

We explain it now. It was for scheduling reasons (those 2 professionals couldn't attend the focus groups)

Results

Consider revising your figure to include the "exceptions to your theory.

They are included now

Line 280-HT was never defined.

We changed this

In general, it feels that the authors endorse that chronic illness is leading to depression, can this please be elaborated in and provide more examples from the literature that support this.

We think section 4.1 now has enough examples from the literature to support this possibility (we acknowledge that this doesn't always happen: see the "exceptions" in our model)

Section 3.1.2.3-Do patients with chronic health problems, but no depression, also have mistrust of Healthcare providers? Please include some acknowledgement of this as either providing data or as a limitation of the study.

We included this. Thank you